# Improving Language Models via Plug-and-Play Retrieval Feedback

## Abstract

Large language models (LLMs) exhibit remarkable performance across various NLP tasks. However, they often generate incorrect or hallucinated information, which hinders their practical applicability in real-world scenarios. Human feedback has been shown to effectively enhance the factuality and quality of the generated content, addressing some of these limitations. However, this approach is resource-intensive, involving substantial manual inputs, which can be time-consuming and expensive. Moreover, human feedback is hard to collect on-the-fly during inference, further limiting its practical utility in dynamic and interactive scenarios. In this paper, we introduce REFEED, a novel pipeline of providing LLMs with automatic retrieval feedback in a plug-and-play manner, without the need of expensive fine-tuning. REFEED first generates initial outputs, then utilizes a retrieval model to acquire relevant information from large document collections. The retrieved information is incorporated into the in-context demonstration to refine the initial outputs, which is more efficient and cost-effective than human feedback or fine-tuning. Experiments on four knowledge-intensive benchmark datasets demonstrate our proposed REFEED could relatively improve 25.7% under zero-shot and 13.5% under few-shot setting, compared to baselines without using retrieval feedback.

## 1 Introduction

Large language models (LLMs) have demonstrated exceptional performance in various NLP tasks, utilizing in-context learning to eliminate the need for task-specific fine-tuning (Brown et al., 2020; Chowdhery et al., 2022; OpenAI, 2023). Such models are typically trained on massive corpora, capturing a wealth of world or domain-specific knowledge within their parameters.

Despite these achievements, LLMs exhibit certain shortcomings, particularly when confronted with complex reasoning and knowledge-intensive tasks (Zhang et al., 2023; Yu et al., 2023). One prominent drawback is their propensity to hallucinate content, generating information not grounded by world knowledge, leading to untrustworthy outputs and a diminished capacity to provide accurate information (Manakul et al., 2023; Alkaissi & McFarlane, 2023). Another limitation of LLMs is the quality and extent of the knowledge they store. The knowledge embedded within an LLM may be incomplete or out-of-date, as it hinges on the veracity and contemporaneity of the sources in the pre-training corpus (Lazaridou et al., 2022; Shi et al., 2023). Moreover, LLMs cannot "memorize" all world information, especially struggling with the long tail of knowledge from their training corpus (Mallen et al., 2022; Kandpal et al., 2022).

Existing methods for enhancing the factuality of language models involve soliciting human annotators to render feedback on language model outputs, followed by reinforcement learning-based fine-tuning (Nakano et al., 2021; Campos & Shern, 2022; Ouyang et al., 2022; Liu et al., 2023a). While this approach simulates human-to-human task learning environments, it can be exceedingly costly as the size of LLMs is growing exponentially and it requires dedicated feedback from human annotators. Furthermore, once the LLMs are fine-tuned, it is hard to receive real-time human feedback during inference to perform immediate error correction.

---

* All source codes and data will be made publicly available after the conference peer-review process.

In this paper, we aim to provide automatic feedback in a plug-and-play manner without the need for fine-tuning LLMs. We explore *two primary research questions*: First of all, can we employ a retrieval module to provide feedback on individual generated outputs without relying on human annotators? Second, can we integrate the retrieval feedback to refine previous generated outputs in a plug-and-play manner, circumventing the expensive fine-tuning of language models? With regards to the two questions posed, we propose a novel pipeline for improving language model inference through automatic retrieval feedback, named REFEED, in a plug-and-play manner. Specifically, the language model first generates initial outputs, followed by a retrieval model fusing the original query and generated outputs into a new query to retrieve relevant information from large document collections such as Wikipedia. The retrieved information enables the language model to reconsider the initial outputs, and optionally refine them to generate new answers.

Notably, compared to retrieve-then-read methods (Lewis et al., 2020; Lazaridou et al., 2022; Shi et al., 2023), REFEED capitalizes on the improved relevance of the retrieved documents, providing either supportive or counteractive evidence to the initial output it generates. Without fusing the initial output into the retrieval query, the document is hard to be retrieved due to the lexical and semantic gap between it and the original question. We discuss the detailed comparison in related work.

To further enhance our proposed REFEED pipeline, we introduce two innovative modules within this framework. Firstly, we diversify the initial generation step by sampling multiple output candidates, leading to a more diverse set of retrieved documents which improves answer coverage. Secondly, we employ an ensemble approach on the outputs before and after retrieval feedback using a probability-based ranking method, as the retrieval feedback may occasionally mislead the language model, where the challenge is also pointed out in Chen et al. (2023).

Overall, the main contributions our paper are:

1. A novel pipeline using retrieval feedback to improve LLMs in a plug-and-play manner.

2. Two advanced modules to further improve the proposed pipeline, specifically diversifying the initial generation outputs and ensembling initial and post-feedback outputs.

3. State-of-the-art performance on three challenging knowledge-intensive tasks under the both zero-shot and few-shot setting.

## 2 RELATED WORK

### 2.1 SOLVING KNOWLEDGE-INTENSIVE TASKS VIA RETRIEVE-THEN-READ PIPELINE.

Mainstream methods for solving knowledge-intensive tasks follows a *retrieve-then-read* paradigm. Given an input query, a retriever is employed to search a large evidence corpus (e.g., Wikipedia) for relevant documents that may contain the answer. Subsequently, a reader is used to scrutinize the retrieved documents and predict an answer. Recent research has primarily focused on improving either the retriever (Karpukhin et al., 2020; Qu et al., 2021; Sachan et al., 2022) or the reader (Izacard & Grave, 2021), as well as training the entire system end-to-end (Singh et al., 2021; Shi et al., 2023). Compared to *retrieve-then-read* pipelines like RePLUG (Shi et al., 2023), our method benefits from the improved relevance of the retrieved documents that elucidate the relationship between query and outputs. Without fusing the initial output into the retrieval query, the text supporting the output cannot be easily identified due to the lack of lexical and semantic overlap with the question.

### 2.2 ALIGNING LANGUAGE MODEL WITH INSTRUCTIONS VIA HUMAN FEEDBACK.

Human feedback plays a crucial role in evaluating language model performance, addressing accuracy, fairness, and bias issues, and offering insights for model improvement to better align with human expectations. Recognizing the significance of integrating human feedback into language models, researchers have developed and tested various human-in-the-loop methodologies (Nakano et al., 2021; Campos & Shern, 2022; Ouyang et al., 2022; Liu et al., 2023a; Scheurer et al., 2023). Instruct-GPT (Ouyang et al., 2022) was a trailblazer in this domain, utilizing reinforcement learning from human feedback to fine-tune GPT-3 to adhere to a wide range of instructions. It trained a reward model to predict the preferred model output based on the feedback from human annotators. The reward model is then used to further fine-tune GPT-3 via Proximal Policy Optimization (PPO).

| Models | Target domain | Diverse Feedback | Ensemble (before & after feedback) | Efficiency |
|---|---|---|---|---|
| Rethinking Retrieval (He et al., 2023) | Commonsense | No | No | High |
| LLM-AUGMENTER (Peng et al., 2023) | Open-domain | No | No | Low |
| REFEED (our proposed method) | Open-domain | Yes | Yes | High |

Table 1: A qualitative comparison with contemporary research reveals distinct advantages of our approach. Although none of the aforementioned works, including our own, has been published in conferences or journals yet, there are clear contrasts to highlight when juxtaposed with Peng et al. (2023) work. Specifically, our research posits that retrieved documents can be directly harnessed as feedback to enhance language model outputs, thereby increasing efficiency markedly. Building on this cornerstone of retrieval feedback, we also debut two innovative modules: the diversification of retrieval feedback and the ensemble of both initial and post-feedback outputs. In comparison to existing research, our proposed REFEED offers a distinctive contribution to the ongoing discourse.

While this methodology adeptly replicates human-to-human task learning paradigms, the fine-tuning of LLMs is remarkably resource-intensive due to the exponential increase in LLM dimensions and the imperative for extensive annotator feedback. Moreover, after fine-tuning, LLMs lack the capacity for real-time human feedback integration during inference or immediate error rectification.

To summarize, our REFEED methodology stands apart in the retrieval-augmented generation (RAG) pipeline (Lewis et al., 2020; Izacard et al., 2022). Unlike conventional approaches that directly use retrieval to enhance model performance in complex reasoning and factual accuracy, our research demonstrates a novel application. We show that retrieved documents can be effectively utilized as feedback to refine language model outputs, significantly boosting efficiency. Our work also diverges from recent advancements such as Chain-of-Thought (Wei et al., 2022), Tree-of-Thought (Yao et al., 2023) reasoning, and Self-Refinement Madaan et al. (2023). These methods do not leverage external knowledge to improve language model reasoning and factual accuracy. Instead, they focus on enhancing the language model's reasoning capabilities through various prompt designs. In comparison to existing research, our proposed REFEED methodology offers a unique and significant contribution to the evolving field of language model development and application.

## 3 PROPOSED METHOD

In this section, we first provide an in-depth description of our innovative plug-and-play retrieval feedback (REFEED) in §3.2.1. The pipeline operates by initially prompting a language model (e.g., InstructGPT) to generate an answer in response to a given query, followed by the retrieval of documents from extensive document collections, such as Wikipedia. Subsequently, the pipeline refines the initial answer by incorporating the information gleaned from the retrieved documents. Then, we introduce two novel modules based on REFEED in §3.2.2. The first module aims to diversify the initial generation step, producing multiple output candidates. This enables the model to identify the most reliable answer by examining the broad range of retrieved documents. The second module employs an ensemble approach that combines language model outputs from both before and after the retrieval feedback process. This is achieved using a probability-based ranking method, which mitigates the risk of retrieval feedback inadvertently misleading the language model.

### 3.1 BACKGROUND.

Traditional large language models, such as GPT-3.5 based architectures, have primarily focused on encoding an input $x$ (e.g., a user query) and predicting the corresponding output output $y$ (Brown et al., 2020; Ouyang et al., 2022). This can be represented as $p(y|x; \theta)$, where $\theta$ denotes the pre-trained model parameters. However, this direct approach often leads to suboptimal performance, because it does not fully exploit the wealth of supplementary world knowledge available to the model (Levine et al., 2022). To address this limitation, recent research has explored methods to improve model performance by incorporating an additional auxiliary variable, corresponding to a *retrieved document* ($d$). This extension modifies the model formulation to $p(y|x) = \sum_i p(y|d_i, x)p(d_i|x)$, marginalizing over all possible documents. We assume, w.l.o.g., that these documents are $d_1, \ldots, d_k$,

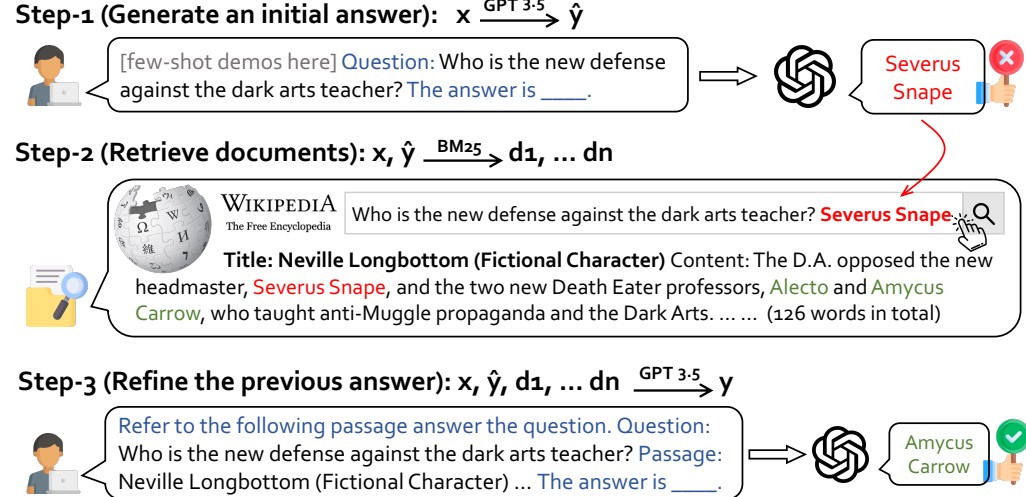

Figure 1: REFEED operates by initially prompting a language model to generate an output to a given query [STEP-1], followed by the retrieval of documents from extensive document collections [STEP-2]. Subsequently, the pipeline refines the initial output by incorporating the information gleaned from the retrieved documents [STEP-3].

yielding $p(y|x) = \sum_{i=1}^{k} p(y|d_i, x)p(d_i|x)$. This technique is referred to as the *retrieve-then-read* pipeline (Lazaridou et al., 2022; Shi et al., 2023).

## 3.2 PROPOSED METHOD: REFEED

### 3.2.1 BASIC PIPELINE

Contrary to traditional methods mentioned above, REFEED is designed to offer feedback via retrieval targeted specifically to individually generated outputs. It can be formulated as $p(y|x) = \sum_i p(y|d_i, x, \widehat{y})p(d_i|\widehat{y}, x)p(\widehat{y}|x)$, where $\widehat{y}$ represents the initial output, $y$ is the final output, and $d_i$ is conditioned not only on $x$ but also on $\widehat{y}$. Thus, $d_i$ is intended to provide feedback specifically on $\widehat{y}$ as the output, rather than providing general information to the query $x$. As in the case of the *retrieve-then-read* pipeline, we retain only the top $k = 10$ highest ranked documents: $p(y|x) = \sum_{i=1}^{k} p(y|d_i, x, \widehat{y})p(d_i|\widehat{y}, x)p(\widehat{y}|x)$.

This method enables a smooth integration of feedback to refine initial outputs in a plug-and-play fashion, eliminating the need for costly fine-tuning. Essentially, REFEED first prompts a language model to produce an initial output, followed by the retrieval of documents from external sources where the initial output is fused into the retrieval query. Then, the initial output is refined as the model incorporates the information from the retrieved documents. In this way, REFEED capitalizes on the improved relevance of retrieved documents, providing either supportive or counteractive evidence to the initial output it generates. The three-step pipeline is illustrated in Figure 1 and outlined below.

**STEP-1: Generate an Initial Output.** In this step, our primary objective is to prompt a language model to generate an output based on the given question. Various decoding strategies can be employed, and we opted for *greedy decoding* due to its simplicity and reproducibility. This step is essential for establishing a foundation upon which the following steps can build and refine the initial output.

**STEP-2: Retrieve Supporting Documents.** The second step in our pipeline involves utilizing a retrieval model (e.g., BM25) to acquire a set of document from an extensive document collection, such as Wikipedia. The initial output $\widehat{y}$ is concatenated with the original question as the retrieval query to fill the lexical and semantic gap between the question and supporting documents. The primary goal of this step is to identify relevant information that can either corroborate or challenge the connection inferred between the question and the initially generated output.

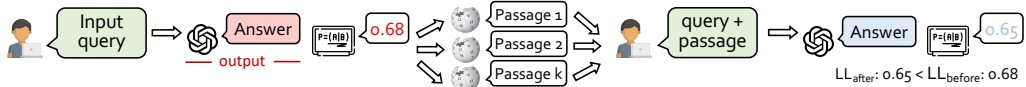

Figure 2: Rather than generating only one initial output, we prompt the language model to sample multiple outputs, allowing for a more comprehensive retrieval feedback based on different outputs.

Figure 3: Instead of directly outputting the refined output, we employ an ensemble method that assesses both the initial and refined answers, thereby allowing for a re-evaluation of answer trustworthiness. In the example depicted in the figure, the language modeling probability is $0.68$ before retrieval feedback and $0.65$ after. Thus, the model selects the initial answer as the final output.

**STEP-3: Refine the Previous Output.** The final step of our pipeline focuses on refining the previously generated output by taking into account the document retrieved in [STEP-2]. During this stage, the language model evaluates the retrieved information and adjusts the initial output accordingly, ensuring that the final output matches with the information in the retrieved documents. This refinement process may involve rephrasing, expanding, or even changing the output based on the newfound knowledge (though it may also choose to keep the original output).

### 3.2.2 ENHANCED MODULES

**MODULE-1: Diversifying Retrieval Feedbacks for Initial Outputs.** Rather than merely generating a single output with the highest probability, we implement sampling methods to produce a set of potential outputs. This approach fosters diversity in the generated outputs and enables a more comprehensive retrieval feedback based on diverse outputs. To elaborate, we feed the input $x$ along with a text prompt into the model, which subsequently samples multiple distinct outputs, denoted as $p(y_j|x;\theta)$. We then utilize the $n$ generated outputs as input queries for the retrieval process, i.e., $[x, y_1], \cdots, [x, y_n]$. This stage is realized by multiple decoding passes, wherein the input query is fed into the language model with nucleus sampling. This strategy increases the probability of obtaining a more diverse set of retrieved documents encompassing a broader spectrum of relevant information. Formally, it can be represented as $p(y|x) = \sum_{i,j} p(y|d_{i,j}, x, \widehat{y}_j) p(d_{i,j}|\widehat{y}_j, x) p(\widehat{y}_j|x)$.

Considering the limitations on the number of documents that can be fed into the language model, we merge all retrieved documents (across different $\widehat{y}_j$), rank them based on query-document similarity scores (given by a retrieval model such as BM25), and retain only the top-$k$ documents for further processing, where $k = 10$ for fair comparisons with baselines. Since documents retrieved from various initial outputs may be duplicated, we merge all retrieved documents and rank them according to their similarity scores from the retrieval model. We then retain only the top-$k$ documents from the entire collection. Lastly, when computing the final output, we provide all $n$ generated outputs as well as the aforementioned top-$k$ documents as part of the prompt. Formally, this can be represented as $p(y|x) = \sum_{i=1}^{k} \sum_{j=1}^{n} p(y|d_{i,j}, x, \widehat{y}_j) p(d_{i,j}|\widehat{y}_j, x) p(\widehat{y}_j|x)$. By incorporating diversity in output generation in [STEP-1], we effectively broaden the potential output space, facilitating the exploration of a wider variety of possible solutions.

**MODULE-2: Ensembling Initial and Post-Feedback Outputs.** Retrieval feedback serves as a crucial component in obtaining relevant information to validate the accuracy of initially generated outputs. Nonetheless, there may be instances where the retrieved documents inadvertently mislead the language model, causing a correct output to be revised into an incorrect one (see examples in Figure 5). To address this challenge, we introduce an ensemble technique that considers both the initial output and the refined output, ultimately improving the overall performance.

In ensemble process, we utilize the average language modeling probability to rank the generated outputs before (i.e., $P_{\text{before}}(y|x) = \frac{1}{t} \sum_{i=1}^{t} p(y_i|y_{<i}, x)$) and after incorporating retrieved documents

| Models | NQ | | TriviaQA | | HotpotQA | | WoW | |
|---|---|---|---|---|---|---|---|---|
| | EM | F1 | EM | F1 | EM | F1 | F1 | R-L |
| *close book methods without using retriever* | | | | | | | | |
| QA prompt (Text-Davinci-003) | 29.9 | 35.4 | 65.8 | 73.2 | 26.0 | 28.2 | 14.2 | 13.3 |
| GenRead (Yu et al., 2023) | 32.5 | 42.0 | 66.2 | 73.9 | 36.4 | 39.9 | 14.7 | 13.5 |
| Self-Prompting (Li et al., 2023) | 36.2 | 46.7 | 66.8 | 74.8 | - | - | - | - |
| *open book methods with using retriever* | | | | | | | | |
| Retrieve-Read (Lazaridou et al., 2022) | 31.7 | 41.2 | 61.4 | 67.4 | 35.2 | 38.0 | 14.6 | 13.4 |
| RePLUG (Shi et al., 2023) | 34.7 | 44.5 | 66.5 | 74.0 | 37.5 | 41.2 | - | - |
| **REFEED (Our method)** | **39.6** | **48.0** | **68.9** | **75.2** | **41.5** | **45.1** | **15.1** | **14.0** |

Table 2: REFEED achieves SoTA performance on three zero-shot knowledge intensive NLP tasks, spanning across four benchmark datasets. The backbone model is text-davinci-003, which is fine-tuned to follow human instructions under zero-shot setting (Ouyang et al., 2022).

| Models | NQ | | TriviaQA | | HotpotQA | | WoW | |
|---|---|---|---|---|---|---|---|---|
| | EM | F1 | EM | F1 | EM | F1 | F1 | R-L |
| Backbone Language Model: Text-Davinci-003 | | | | | | | | |
| *close book methods without using retriever* | | | | | | | | |
| QA prompt (Text-Davinci-003) | 36.5 | 46.3 | 71.2 | 76.5 | 31.2 | 37.5 | 14.1 | 13.3 |
| GenRead (Yu et al., 2023) | 38.2 | 47.3 | 71.4 | 76.8 | 36.6 | 47.5 | 14.7 | 14.1 |
| *open book methods with using retriever* | | | | | | | | |
| Retrieve-Read (Lazaridou et al., 2022) | 34.3 | 45.6 | 66.5 | 70.6 | 35.2 | 46.8 | 14.5 | 13.8 |
| RePLUG (Shi et al., 2023) | 36.7 | 46.3 | 69.8 | 74.0 | 36.0 | 47.2 | - | - |
| **REFEED (Our method)** | **40.1** | **50.0** | **71.8** | **77.2** | **41.5** | **54.2** | **15.1** | **14.3** |
| Backbone Language Model: Code-Davinci-002 (Codex) | | | | | | | | |
| *close book methods without using retriever* | | | | | | | | |
| QA prompt (Codex) | 41.6 | 52.8 | 73.3 | 79.2 | 32.5 | 42.8 | 16.9 | 14.7 |
| GenRead (Yu et al., 2023) | 44.2 | 55.2 | 73.7 | 79.6 | 37.5 | 48.8 | 17.2 | 15.1 |
| *open book methods with using retriever* | | | | | | | | |
| Retrieve-Read (Lazaridou et al., 2022) | 43.9 | 54.9 | 75.5 | 81.7 | 41.5 | 53.7 | 17.0 | 14.9 |
| RePLUG (Shi et al., 2023) | 44.6 | 55.0 | 75.6 | 81.7 | 42.0 | 54.5 | - | - |
| **REFEED (Our method)** | **46.4** | **57.0** | **76.6** | **82.7** | **43.5** | **56.5** | **17.6** | **15.5** |

Table 3: REFEED achieved SoTA performance in three few-shot knowledge intensive NLP tasks. Besides text-davinci-003, codex was evaluated, given its demonstrated prowess in few-shot settings.

(i.e., $P_{\text{after}}(y|x) = \frac{1}{t} \sum_{i=1}^{t} p(y_i|y_{<i}, x, \widehat{y}, d)$). If the probability of an output before retrieval feedback is higher than that after retrieval feedback, we retain the initially generated output, otherwise we choose the refined output. This strategy allows for a more informed assessment of the trustworthiness of output before and after retrieval feedback, ensuring a more accurate final response.

## 4 EXPERIMENTS

In this section, we conduct comprehensive experiments on three knowledge-intensive NLP tasks, including single-hop QA (i.e., NQ (Kwiatkowski et al., 2019), TriviaQA (Joshi et al., 2017)), multi-hop QA (i.e., HotpotQA (Yang et al., 2018)) and dialogue generation (i.e., WoW (Dinan et al., 2019)). In single-hop QA datasets, we employ the same splits (i.e., unfiltered) as Karpukhin et al. (2020); Izacard & Grave (2021). With regard to the HotpotQA and WoW datasets, we use the split from the KILT challenge (Petroni et al., 2021). More detailed experimental settings can be found in Table A.1 in Appendix. Besides, the hyper-parameter setting can be found in Section A.1 in Appendix.

Besides, we also incorporate Recall@K (R@K) as an intermediate evaluation metric, which is calculated as the percentage of top-K retrieved or generated documents containing the correct answer (Karpukhin et al., 2020). When evaluating open-domain dialogue systems, we adhere to the guidelines set forth by the KILT benchmark (Petroni et al., 2021), which recommends using a combination of F1 and Rouge-L (R-L) scores as evaluation metrics.

| Models | NQ | | TriviaQA | | HotpotQA | | WoW | |
|---|---|---|---|---|---|---|---|---|
| | EM | F1 | EM | F1 | EM | F1 | F1 | R-L |
| **REFEED (Our method)** | **46.4** | **57.0** | **76.6** | **82.7** | **43.5** | **56.5** | **17.6** | **15.5** |
| ⊢ w/o diverse retrieval feedback | 45.1 | 56.2 | 75.9 | 82.1 | 42.1 | 54.8 | 17.0 | 14.8 |
| ⊢ w/o ensemble before & after | 45.5 | 56.5 | 76.1 | 82.4 | 42.5 | 55.3 | 17.1 | 14.9 |

Table 4: Ablation Study. Our ensemble method and diversifying generation in REFEED can improve model performance on four benchmark datasets. The backbone model is code-davinci-002 (codex).

## 4.1 BASELINE METHODS

In our comparative analysis, we assess our proposed model against two distinct groups of baseline methodologies. The first group encompasses closed-book models, including InstructGPT (Ouyang et al., 2022), GenRead (Yu et al., 2023) and Self-Prompting (Li et al., 2023), which operate without the assistance of any external supporting documents. Each of these baseline methods adheres to a uniform input format, specifically utilizing the structure: [prompt words; question]. More details about prompt choices can be found in Table 10 and Table 11 in Appendix.

The second group of models adheres to a *retrieve-read* pipeline (Lazaridou et al., 2022; Shi et al., 2023), which entails a two-stage process. In the initial stage, a retriever component is employed to identify and extract a select number of relevant documents pertaining to a given question from an extensive corpus, such as Wikipedia. Subsequently, a reader component is tasked with inferring a conclusive answer based on the content gleaned from the retrieved documents. All baseline methods within this the group adhere to a standardized input format, which is defined as: [prompt words; passage; question]. **We note that** we did not evaluate the RePLUG on the WoW dataset due to its reliance on ensemble methods, which require significantly more tokens for long-text generation.

## 4.2 EXPERIMENTAL ANALYSIS

### 4.2.1 ZERO/FEW-SHOT QUESTION ANSWERING AND DIALOGUE EVALUATION

In zero-shot setting, there is no training question-answer pairs and conversational input-output pairs as demonstrations for the models. Consequently, all models are expected to generate answers solely based on the input test question provided.

For the purposes of our experiments, we utilized text-davinci-003 as the backbone model due to its remarkable performance in zero-shot scenarios. As shown in Table 2, REFEED outperforms baseline methods by effectively leveraging retrieval feedback. In particular, REFEED exhibits a significant improvement in EM scores by +7.7 on two open-domain QA benchmarks in comparison to the original text-davinci-003. We also observe a similar trend in the context of multi-hop QA tasks and dialogue systems, where our proposed REFEED consistently surpasses the baseline model.

Moreover, when juxtaposed with methods that directly retrieve or generate documents, REFEED demonstrates a markedly superior performance. This can be attributed to the fact that alternative methods often struggle to retrieve relevant passages when there is an absence of lexical and semantic overlap between the query and the documents to retrieve. On the other hand, our proposed REFEED offers a more robust and accurate retrieval solution for knowledge-intensive tasks by utilizing the initial generated output to fill this gap.

In the few-shot setting, as shown in Table 3, we observed a similar pattern to the zero-shot setting, further reinforcing the effectiveness of our method. This consistency across various settings underscores the model's versatility and adaptability, illustrating its potential to deliver superior results across a wide range of question-answering and dialogue evaluation tasks.

### 4.2.2 ABLATION STUDY ON ENSEMBLE METHOD AND DIVERSE GENERATION

**MODULE-1: Diverse Retrieval Feedback.** As shown in Table 4, the performance of REFEED declines by an average of 1.1 EM score across three QA datasets when diverse generation is not utilized. This observation underscores the significance of incorporating diverse generation, as it can lead to multiple, distinct answers, leading to a more diverse set of documents retrieved during subsequent stages. This further leads to a positive improvement on the answer hit ratio among the

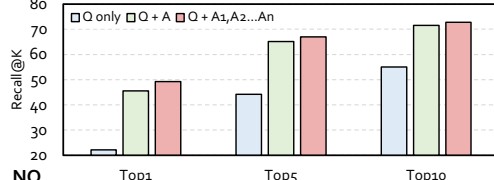 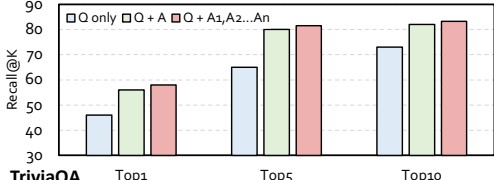

Figure 4: Recall@K on test sets, measured as the percentage of top-K documents containing correct answer. The "Q only" refers to direct retrieval based on the input query, where the "Q + A" represents generating only one initial answer, and the "Q + A1,A2...An" represents *n* diverse answers in STEP-1.

| Models | ChatGPT3.5 | | | | | | GPT-4 | | | | | |
|---|---|---|---|---|---|---|---|---|---|---|---|---|
| | NQ | | TriviaQA | | HotpotQA | | NQ | | TriviaQA | | HotpotQA | |
| | EM | F1 | EM | F1 | EM | F1 | EM | F1 | EM | F1 | EM | F1 |
| QA prompt | 32.3 | 39.9 | 65.6 | 69.5 | 23.5 | 24.0 | 34.8 | 49.6 | 64.6 | 72.8 | 30.8 | 33.7 |
| Retrieve-Read | 34.3 | 41.5 | 58.7 | 63.7 | 31.7 | 33.6 | 32.5 | 46.5 | 59.9 | 67.1 | 31.6 | 37.5 |
| **REFEED** | 37.5 | 48.1 | 66.3 | 71.1 | 34.1 | 36.0 | 36.8 | 54.4 | 66.3 | 74.0 | 36.9 | 42.6 |

Table 6: REFEED consistently outperforms baseline methods when using ChatGPT3.5 and GPT-4.

retrieved documents, as shown in Figure 4, which is a consistent finding with Wang et al. (2023). The increased evidence coverage improves the overall quality and relevance of the information obtained during retrieval, and consequently improves the final answer accuracy.

**MODULE-2: Ensemble before & after Feedback.** As shown in Table 4, it is evident that the performance of REFEED declines by 0.8 EM score across three QA datasets when the ensemble method is not employed. This finding highlights the importance of implementing an ensemble strategy before and after retrieval feedback. The ensemble method effectively utilizes the language model's inherent beliefs (i.e., knowledge stored in model parameters) in conjunction with the retrieval feedback, opting for the more likely answer between the initial and refined ones, thus mitigating the possible negative impact of the retrieved documents on the model's performance.

### 4.2.3 ANALYSIS ON CHAIN-OF-THOUGHT REASONING ON MULTI-HOP QA

In this section, we study the compatibility of our pipeline with more advanced prompting methods like chain-of-thought reasoning (CoT) (Wei et al., 2022). CoT entails the generation of a sequence of intermediate reasoning steps before reaching the final answer. With CoT, the model can significantly enhance its performance in complex reasoning scenarios, such as multi-hop reasoning tasks, as shown in Table 8. We implemented REFEED in conjunction with CoT reasoning by generating intermediate reasoning steps prior to arriving at the final answer. Following this, we utilized the answer to retrieve

| Models | HotpotQA | |
|---|---|---|
| | EM | F1 |
| *close book methods without using retriever* | | |
| QA Prompt (Brown et al., 2020) | 32.5 | 42.8 |
| CoT Prompt (Wei et al., 2022) | 35.0 | 46.8 |
| MCR Prompt (Yoran et al., 2023) | - | 57.0 |
| *open book methods with using retriever* | | |
| Retrieve-Read with CoT Prompt | 42.1 | 54.8 |
| REFEED with CoT Prompt | **44.2** | **57.4** |

Table 5: REFEED can be applied to chain-of-thought (CoT) reader as well, on multi-step reasoning task. The backbone model is Codex.

documents for feedback and subsequently generated another CoT reasoning to refine the previously generated outputs. This approach led to a significant improvement on complex QA scenarios in the HotpotQA, when compared to employing straightforward QA prompts, including advanced promting methods like CoT prompting and meta-reasoning over multiple chains of thought (MCR) (Yoran et al., 2023). To summarize, our proposed REFEED can be seamlessly integrated with CoT reasoning, thereby showcasing their complementary nature. The successful combination of REFEED and CoT enables the model to handle more intricate tasks and exhibits its potential for tackling real-world challenges that demand complex problem-solving capabilities.

---

**Question:** When was the Deadpool 2 movie being released?     **Gold answer: May 18, 2018**
**Retrieved document (Q only):** Deadpool 2 (ID 18960867) ... Also in April, Leslie Uggams confirmed that she would be reprising her role of Blind Al from the first film, while Fox gave the sequel a `June 1, 2018` release date. ...
**Retrieved document (Q + generated A):** Deadpool 2 (ID 18960900) ... "Deadpool 2" was released in the United States on `May 18, 2018`, having been previously scheduled for release on `June 1` of that year. Leitch's initial cut of the film was around two hours and twelve minutes, with "nips and tucks" done to it to get the run time down to ...
**Text-davinci-003:** June 1, 2018 ⊗     **Retrieve-read:** June 1, 2018 ⊗     **Retrieve-Feedback:** May 18, 2018 ✓

- - - - - - - - - - - - - - - - - - - - - - - - - - - - - - - - - - - - - - - - - -

**Question:** When is season 3 of Grace and Frankie being released?     **Gold answer: March 24, 2017**
**Retrieved document (Q only):** Grace and Frankie (ID 18251210) ... On December 10, 2016, the series was renewed for a third season which premiered on `March 24, 2017`. On April 12, 2017, the series was renewed for a fourth season, which premiered on `January 19, 2018`. ...
**Retrieved document (Q + generated A):** Grace and Frankie (ID 18251208) It premiered on Netflix on May 8, 2015, with all 13 episodes of the first season released simultaneously. The second, third, and fourth seasons, also consisting of 13 episodes each, have been released on May 6, 2016, `March 24, 2017`, and `January 19, 2018`.
**Text-davinci-003:** March 24, 2017 ✓  **Retrieve-read:** March 24, 2017 ✓  **Retrieve-Feedback:** January 19, 2018 ⊗

Figure 5: Case Studies. The first instance shows the language model effectively refines answers into accurate ones by utilizing retrieval feedback. On the contrary, the second instance demonstrates the response is misguided by the documents after retrieval, culminating in an inaccurate response.

### 4.2.4 EXPERIMENTS ON MOST RECENT CHATGPT / GPT-4

As illustrated in Table 6, our REFEED method demonstrates consistent improvements in QA performance. This enhancement is evident when compared to both the standard closed-book QA method and the retrieve-then-read pipeline. Furthermore, it's noteworthy that while the traditional retrieve-then-read system is sometimes negatively impacted by noisy retrieval, our ReFeed approach effectively circumvents this issue. This advantage is particularly apparent if the initial round of generation yields accurate results. By improving robustness against retrieval noise, REFEED not only enhances accuracy but also ensures a more reliable and stable performance under varied conditions.

## 5 CONCLUSION

In conclusion, this paper presents a novel pipeline, REFEED, designed to improve large language model in a plug-and-play framework. By employing a retrieval method to provide automatic feedback on generated outputs and integrating this feedback to refine the outputs, REFEED offers a practical and efficient solution without the need for expensive fine-tuning. We further introduce two innovative modules within the REFEED pipeline: diverse answer generation and an ensemble approach. These two modules further enhance REFEED to produce more reliable and accurate answers by considering a wider array of retrieved documents and mitigating the risk of misleading retrieval feedback. Our extensive experiments on four challenging knowledge-intensive benchmarks demonstrate the effectiveness of REFEED under zero and few-shot settings. We believe the retrieval feedback idea has the potential to be widely adopted in a variety of scenarios and applications in the future.

## LIMITATION

Despite the significant contributions of this work, it is important to acknowledge following limitations:

This model aims to enhance natural language generation tasks, utilizing document retrieval as feedback to refine the content generated by language models. As such, its optimal use is seen in tasks necessitating the creation of textual content. However, its specialized design may limit its flexibility in handling language understanding tasks, such as binary classification, which demand discrete outputs such as 'yes' or 'no', might not be well supported by the model's architecture. This constraint arises from the model's inherent requirement for the initially generated content to be narrative text.

Besides, our approach leverages an ensemble method, combining initial and post-feedback outputs via comparing the probability. However, this strategy relies on probability, which is contingent upon the calibration of the language model. We acknowledge that for optimal performance, our approach necessitates a language model that is relatively well-calibrated. In scenarios where the model's calibration is less than ideal, the effectiveness of our ensemble approach may be compromised.

REPRODUCIBILITY STATEMENT

To ensure the reproducibility of our experiments and benefit the research community, we will open-source all source codes and data after the conference peer-review process. The hyper-parameters and other variable required to reproduce our experiments are described in Table 7. All large language models used in our paper are publicly available. We note that reproducing experiments with GPT-3 series models requires access to the GPT-3 API provided by OpenAI.

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

# A   APPENDIX

## A.1   DATASET INTRODUCTION

– **TriviaQA** (Joshi et al., 2017). This dataset is used for the evaluation and training of question-answering models. It contains trivia questions along with their answers, which were collected from trivia and quiz-league websites. The dataset is designed to simulate the kinds of questions that might be asked in a trivia game, and can be used to train models that can answer general questions.

– **Natural Questions (NQ)** (Kwiatkowski et al., 2019). This dataset is constructed from real user queries issued to the Google search engine. Human annotators paired these questions with short and long answers found in Wikipedia. This dataset can be used for training question-answering models to provide concise answers to user queries, and is particularly geared towards questions that people actually ask search engines.

– **HotpotQA** (Yang et al., 2018). This dataset focuses on multi-hop question answering. In this dataset, the questions are designed in such a way that the model must gather information from several different parts of a document or across different documents to arrive at the correct answer. This dataset can be used to train and evaluate models that are capable of complex reasoning over text.

– **Wizard of Wikipedia (WoW)** (Dinan et al., 2019). This is an open-domain dialogue task for training agents that can converse knowledgeably about open-domain topics. One speaker in the conversation must ground their utterances in a specific knowledge sentence from a Wikipedia page.

## A.2   BACKBONE LANGUAGE MODEL

**Codex**: OpenAI Codex, i.e., code-davinci-002, a sophisticated successor to the GPT-3 model, has undergone extensive training utilizing an immense quantity of data. This data comprises not only natural language but also billions of lines of source code obtained from publicly accessible repositories, such as those found on GitHub. As a result, the Codex model boasts unparalleled proficiency in generating human-like language and understanding diverse programming languages.

**Text-davinci-003:** Building on the foundation laid by previous InstructGPT models, OpenAI's text-davinci-003 represents a significant advancement in the series. This cutting-edge model showcases considerable progress in multiple areas, including the ability to generate superior quality written content, an enhanced capacity to process and execute complex instructions, and an expanded capability to create coherent, long-form narratives.

After careful consideration, we ultimately decided against employing ChatGPT and GPT-4 as the backbone language models for our project. The primary reason for this decision is OpenAI's announcement that both models will be subject to ongoing updates in their model parameters[1]. These continual modifications would lead to non-reproducible experiments, potentially compromising the reliability of our research outcomes.

## A.3   HYPERPARAMETER SETTING

In our experiments, we primarily adjusted specific hyperparameters to control the maximum length of outputs and ensure the stability of the generated outputs. Outside of the diversifying feedback setting, it is imperative that the model produces deterministic outputs to ensure reproducibility of the experiments. Furthermore, as highlighted both in Si et al. (2022) and our preliminary experiments, we observed that the greedy search method (i.e., temperature = 0) outperforms sampling-based methods when generating a singular output. The detailed hyperparameter setting is shown in Table 7.

---

[1] https://platform.openai.com/docs/models/gpt-3-5

| Methods | Maximum output | Temperature | Top_P |
|---|---|---|---|
| QA prompt (Brown et al., 2020) | 20 | 0.0 | 0.0 |
| Retrieve-read (Lazaridou et al., 2022) | 20 | 0.0 | 0.0 |
| REFEED (our method) | 20 | 0.0 | 0.0 |
| ⊢ with diverse retrieval feedback | 20 | 0.7 | 0.9 |

Table 7: Hyperparaters settings used in the experiments when using text-davinci-003 and codex. Except for diverse retrieval feedback, all other methods utilize the greedy decoding.

| Backbone Model: ChatGPT-3.5-turbo | $k$=1 | | | $k$=5 | | |
|---|---|---|---|---|---|---|
| | NQ | TriviaQA | HotpotQA | NQ | TriviaQA | HotpotQA |
| QA Prompt (Brown et al., 2020) | 32.3 | 65.6 | 23.5 | 32.3 | 65.6 | 23.5 |
| Retrieve-Read (Lazaridou et al., 2022) | 23.6 | 40.5 | 23.2 | 30.5 | 50.4 | 28.5 |
| REFEED (our method) | 34.0 | 65.9 | 27.1 | 34.3 | 67.5 | 30.0 |

| Backbone Model: ChatGPT-3.5-turbo | $k$=10 | | | $k$=20 | | |
|---|---|---|---|---|---|---|
| | NQ | TriviaQA | HotpotQA | NQ | TriviaQA | HotpotQA |
| QA Prompt (Brown et al., 2020) | 32.3 | 65.6 | 23.5 | 32.3 | 65.6 | 23.5 |
| Retrieve-Read (Lazaridou et al., 2022) | 34.3 | 58.7 | 31.7 | 34.6 | 59.5 | 31.8 |
| REFEED (our method) | 37.5 | 66.3 | 34.1 | 37.8 | 66.5 | 34.4 |

Table 9: Comparative performance analysis of varying document retrieval counts ($k$) Using ChatGPT-3.5-turbo on NQ, TriviaQA, and HotpotQA.

## A.4 EFFICIENCY ANALYSIS

In this section, we delve into an intricate comparison of computational costs when employing retrieval feedback, contrasting it against baseline of "without retrieval" and "retrieve-read" pipeline.

– Comparison to the "w/o retrieval" baseline: While our REFEED methodology does entail an increment in computational overhead, the trade-off manifests itself in a substantial performance uptick. Specifically, there is an average enhancement of 4.6% on the NQ dataset. With the continuous decline in inference costs of LLMs and the burgeoning emergence of open-source LLMs, this balance between computational cost and model accuracy will lean progressively towards being more favorable in the near future.

| Methods | Avg. Input | Avg. Output | EM |
|---|---|---|---|
| *close book methods without using retriever* | | | |
| QA Prompt | 876 words | 3.1 words | 41.6 |
| *open book methods with using retriever* | | | |
| Retrieve-Read | 1,875 words | 2.8 words | 43.9 |
| REFEED (ours) | 2,754 words | 2.9 words | 46.4 |

Table 8: Computation cost comparison when employing retrieval feedback, contrasting it against "w/o retrieval" and "retrieve-read" pipeline.

– Comparison to the "retrieve-read" baseline: When juxtaposed with retrieve-read methods, such as RePLUG, REFEED necessitates only a nominal uptick in costs for generating the initial output. However, the strength of the REFEED pipeline lies in its ability to fine-tune model outputs by leveraging external retrieval feedback. This capability translates to an improvement exceeding 2% on the challenging NQ dataset, culminating in state-of-the-art performance on NQ in a few-shot setting.

To further optimize the system regarding the input context length overhead, we employed a implementation trick that only inserts 10 retrieved documents only to the test input, i.e., no retrieved documents for in-context demonstrations, for both baseline and our method, so they are fairly compared. Our evaluations indicate that this modification has an insignificant impact on model performance. However, it dramatically trims down the total number of input tokens, proving to be an efficient strategy.

## A.5 EFFECTS ON SAMPLING DIFFERENT NUMBER OF DOCUMENTS

Previous research has indicated that increasing the number of retrieved documents, denoted as $k$, beyond 10 leads to only marginal improvements, while also contributing to increased complexity (Si et al., 2022; Shi et al., 2023). Additionally, as noted in Liu et al. (2023b), excessively expanding the

context length for Large Language Models (LLMs) can result in the "lost in the middle" issue, where the LLM tends to ignore the middle part of a long text input.

To further investigate the effect of varying $k$, we conducted experiments using ChatGPT (gpt-3.5-turbo) on datasets such as NQ, TriviaQA, and HotpotQA. The experimental results align with the findings presented in Shi et al. (2023) and Si et al. (2022). As illustrated in the table, setting $k$ to 10 yields significantly better performance compared to when $k$ is set to 1 or 5. However, increasing $k$ to 20 only offers marginal improvements over a setting of $k = 10$.

Additionally, as shown in Table 9, we observed that the retrieve-then-read pipeline can be easily misled by noisy retrieved documents. This issue negatively impacts performance on three open-domain QA datasets when only one document is used for retrieval. In contrast, our REFEED approach effectively combines both the internal knowledge of the language model and external knowledge sources to arrive at the final answer. This integration allows for a more robust and accurate response, particularly in scenarios where single-document retrieval proves insufficient.

## A.6 PROMPT CHOICES AND MORE CASE STUDIES

The prompt choices of close-book and open-book settings are shown in Tables 10-11.
More case studies are shown in Table 12.

Table 10: Close-book prompts. The examples are sourced from the first 64 QA pairs in NQ train split.

Question: total number of death row inmates in the us?
The answer is 2,718

Question: big little lies season 2 how many episodes?
The answer is seven

Question: who sang waiting for a girl like you?
The answer is Foreigner

Question: where do you cross the arctic circle in norway?
The answer is Saltfjellet

Question: who is the main character in green eggs and ham?
The answer is Sam - I - am

... (64 shots in total) ...

Question: [input question] (E.g., When was the Deadpool 2 movie being released?)
The answer output from the model.

Table 11: Open-book prompts. The examples are sourced from the first 64 QA pairs in NQ train split. The documents are retrieved by BM25. In total, 10 documents were used for the prompt.

---

Question: total number of death row inmates in the us?
The answer is 2,718

Question: big little lies season 2 how many episodes?
The answer is seven

Question: who sang waiting for a girl like you?
The answer is Foreigner

Question: where do you cross the arctic circle in norway?
The answer is Saltfjellet

Question: who is the main character in green eggs and ham?
The answer is Sam - I - am

... (64 shots in total) ...

Question: [input question] (E.g., When was the Deadpool 2 movie being released?)
Title: [input title] (E.g., Deadpool). Passage: [input passage] (E.g., Screen Rant called it possibly "the best film marketing campaign in the history of cinema". HostGator's Jeremy Jensen attributed the campaign's success to Reynolds, and to Fox for embracing the film's R rating. "Deadpool"'s world premiere was held at the Grand Rex in Paris on February 8, 2016, before its initial theatrical release in Hong Kong the next day. This was followed by releases in 49 other markets over the next few days, including the United States on February 12. The movie was released in several formats, ...)
Title: [input title] (E.g., No Good Deed). Passage: [input passage] (E.g., a movie that begun and ended on its own terms. There was nothing else to say, because we had said it. Instead, a Deadpool scene was shown as a teaser for "Deadpool 2" before "Logan", confirmed to be the scene directed by Leitch in December 2016. It was written by Rhett Reese and Paul Wernick, writers of the "Deadpool" films. After the initial release, Reese quickly clarified that the scene was not intended to be an official trailer for "Deadpool 2", with none of its footage meant to appear in that film, ...)
... (10 documents in total) ...
Title: [input title] Wikipedia title Passage: [input passage] Wikipedia content
Refer to the passages below and answer the following question with just a few words.
The answer is output from the model.

---

Table 12: Case Studies: The results from both cases demonstrate that our REFEED method can deliver accurate answers with the use of retrieval feedback. Without this feedback, the answers are incorrect.

---

**Question:** Who is the new defence against the dark arts teacher?

**Gold answer:** Amycus Carrow

**Retrieved document (Q only):** Title: Harry Potter. Passage: Harry meets most of the main characters and gains his two closest friends: Ron Weasley, a fun-loving member of an ancient, large, happy, but poor wizarding family, and Hermione Granger, a gifted, bright, and hardworking witch of non-magical parentage. Harry also encounters the school's potions master, Severus Snape, who displays a conspicuously deep and abiding dislike for him, the rich brat Draco Malfoy whom he quickly makes enemies with, and the Defence Against the Dark Arts teacher.

**Retrieved document (Q + generated A):** Title: Neville Longbottom (Fictional Character). Passage: The D.A. opposed the new headmaster, Severus Snape, and the two new Death Eater professors, Alecto and Amycus Carrow, who taught anti-Mugglepropaganda and the Dark Arts. Neville got in trouble with the Carrows for refusing to practise the Cruciatus Curse on other students as a method of punishment, as well as for standing up against their bigotry and cruelty.

**Text-davinci-003:** Severus Snape (✗)

**Retrieve-read:** Severus Snape (✗)

**Retrieve-Feedback:** Amycus Carrow (✔)

---

**Question:** Who sings the wire season 5 theme song?

**Gold answer:** Steve Earle

**Retrieved document (Q only):** Title: The Gospel of Gospel. Passage: choir at the church founded by Aretha Franklin's father, the legendary gospel performer Reverend C.L. Franklin, jams with Robert Randolph and the Family Band and duets with Chaka Khan. Interviewees include Ray Charles, Isaac Hayes, Al Green, B.B. King, Chaka Khan, Alexander O'Neal, Edwin Hawkins, Mary Mary and The Blind Boys Of Alabama. The show was commissioned by British network Channel 4, by the network's commissioning editor for religion Elizabeth Clough, alongside a talent contest for gospel performers. The Gospel of Gospel The Gospel Of Gospel is a 2001 TV documentary made by independent production company Lion Television for Channel

**Retrieved document (Q + generated A):** Title: Way Down in the Hole. Passage: Way Down in the Hole "Way Down in the Hole" is a song written by the singer-song writer Tom Waits. It was included on his 1987 album "Franks Wild Years", which was later made into a stage production. The song was used as the theme for HBO "The Wire". A different recording was used each season.

**Text-davinci-003:** Tom Waits (✗)

**Retrieve-read:** The Blind Boys of Alabama (✗)

**Retrieve-Feedback:** Steve Earle (✔)

---

