# OpenReview forum: "Improving Language Models via Plug-and-Play Retrieval Feedback"
_ICLR.cc/2024/Conference — Submitted to ICLR 2024_

### Official Review · Reviewer_fuhX · 2023-10-28

**Soundness:** 2 fair
**Presentation:** 2 fair
**Contribution:** 2 fair
**Rating:** 5
**Confidence:** 5

**Summary:**

This paper proposes a novel pipeline, REFEED, to provide LLMs with automatic retrieval feedback in a plug-and-play manner, without the need of expensive fine-tuning.
REFEED includes advanced modules to improve the proposed pipeline, specifically diversifying the initial generation outputs and ensembling initial and post-feedback outputs.
That is, REFEED first generates initial outputs, then utilizes a retrieval model to acquire relevant information from large document collections.
Then, the retrieved information is incorporated into the in-context demonstration to refine the initial outputs, which is more efficient and cost-effective than human feedback or fine-tuning.

**Strengths:**

* REFEED is simple architecture that improves large language model in a plug-and-play framework.
* This architecture and a retrieval method allows REFEED a practical and efficient solution without the need for expensive fine-tuning.
* To produce more reliable and accurate answers and mitigate the risk of misleading retrieval feedback, REFEED has equipped with two newly introduced modules, diverse answer generation and an ensemble approach.

**Weaknesses:**

* While REFEED looks promising approach to improve large language model in a plug-and-play framework,
its goals are not specifically clarified.
Because of the wide range of goals, it is necessary to compare and discuss with other approaches, such Prompt tuning, Chain-of-Thought (CoT), Tree of Thoughts (ToT), and, Retrieval Augmented Generation (RAG) have not been described.
Figure 1, 2, and 3 can be interpreted as a kind of RAG.
* Backbone language models are limited, and it is difficult to determine whether the effect is due to the approches or the emergent nature of the language model.
* Reproducibility, limitations, and lack of qualitative evaluation do not confirm its validity. Table 1 is inadequate because the settings are not clear.

**Questions:**

* Pease explian why you chose text-davinci-003 and Code-Davinci-002 (Codex) as the backbone language models?
* Can REFEED avoid generating incorrect or hallucinated information?
* See Weaknesses

---

> ### Author Response · Authors · 2023-11-23
> **Response to Reviewer fuhX [1/2]**
>
> Dear Reviewer fuhX,
>
> Thanks very much for your review! Here are the responses to your questions/weakness.
>
> - Q1: Pease explian why you chose text-davinci-003 and Code-Davinci-002 (Codex) as the backbone language models? Backbone language models are limited, and it is difficult to determine whether the effect is due to the approches or the emergent nature of the language model.
>
>
> Thank you for highlighting the importance of backbone model selection. Our initial decision to use Davinci-003 and Codex as backbone models was made to ensure a fair comparison with baseline methods (e.g., GenRead, RePLUG) that we have discussed in the paper.
>
> Besides, the ChatGPT and GPT-4 system might involve many user data for instruction tuning, which could have contamination with many academic benchmark dataset, such as NQ and TriviaQA.
>
> Lastly, in OpenAI’s announcement that both ChatGPT and GPT-4 will be subject to ongoing updates in their model parameters. These continual modifications would lead to non-reproducible experiments, potentially compromising the reliability of our research outcomes.
>
> In response to your concern, we have conducted additional experiments using ChatGPT (gpt-3.5 turbo) and GPT-4, which reinforce the adaptability and effectiveness of our approach across different model architectures. These results are now detailed in the revised version of our paper, providing a comprehensive view of our method's applicability.
>
>
> | Experiment on ChatGPT (gpt-3.5-turbo)                  | NQ          | TriviaQA    | HotpotQA    |
> |--------------------------------------------------------|-------------|-------------|-------------|
> | Closebook question answering (Question -> Answer)      | 32.3 / 39.9 | 65.6 / 69.5 | 23.5 / 24.0 |
> | Retrieve-then-Read (Question -> Retrieval -> Answer)   | 34.3 / 41.5 | 58.7 / 63.7 | 31.7 / 33.6 |
> | ReFeed (Question -> Answer -> Retrieval -> Refinement) | 37.5 / 48.1 | 66.3 / 71.1 | 34.1 / 36.0 |
>
>
> | Experiment on GPT4 (gpt-4)                             | NQ          | TriviaQA    | HotpotQA    |
> |--------------------------------------------------------|-------------|-------------|-------------|
> | Closebook question answering (Question -> Answer)      | 34.8 / 49.6 | 64.6 / 72.8 | 30.8 / 33.7 |
> | Retrieve-then-Read (Question -> Retrieval -> Answer)   | 32.5 / 46.5 | 59.9 / 67.1 | 31.6 / 37.5 |
> | ReFeed (Question -> Answer -> Retrieval -> Refinement) | 36.8 / 54.4 | 66.3 / 74.0 | 36.9 / 42.6 |
>
> From the above table, we can obserbe:
>
> (1) The retrieval-feedback consistently improves the system, compared to both standard closebook QA method, and retrieve-then-read pipeline.
>
> (2) The traditional retrieve-then-read system even sometimes hurt by noisy retrieval. On the contrary, our ReFeed could avoid misled issue if the first-round generation is good.
>
>
> - Q2: Can REFEED avoid generating incorrect or hallucinated information?
>
> Yes, this is the main contribution of ReFeed, avoiding hallucination via using retrieval feedback to validate the correctness of generated output. The evaluations are mainly based on open-domain factual QA, the improvement reveals the ReFeed could greatly reduce the hallucination / improve factuality. Figure 5 in the paper show some cased scenarios.
>
> - Q3: Reproducibility, limitations, and lack of qualitative evaluation do not confirm its validity. Table 1 is inadequate because the settings are not clear.
>
> Thank you for your suggestions. We acknowledge these aspects are crucial for confirming the validity of our research.
>
> Regarding reproducibility, while this was not included in the main paper, comprehensive details have been provided in Appendices A.1 and A.2. Additionally, to further ensure reproducibility, we will make all codes and data publicly available for sure.
>
> Regarding the limitations of our study, we have addressed this by adding a dedicated section on page 9 of our paper. This section aims to provide a transparent overview of the potential constraints and boundaries of our research.
>
> Regarding qualitative evaluation, all methods were evaluated in a few-shot QA/generation setting. Although the domains vary, with focusing on commonsense QA [1] and on open-domain QA [2], methods share the objective of using feedback to improve the performance of large language models like GPT-3. Table 1 mainly reveals the main differences in method design and application domain, without mentioning their implementation details.
>
> [1] Rethinking with retrieval: Faithful large language model inference.
>
> [2] Check your facts and try again: Improving large language models with external knowledge and automated feedback.

---

> ### Author Response · Authors · 2023-11-23
> **Response to Reviewer fuhX [2/2]**
>
> - Q4: Related work discussion on Prompt tuning, Chain-of-Thought (CoT), Tree of Thoughts (ToT), and, Retrieval Augmented Generation (RAG).
>
>
> Thank you for incorporating the related work discussion into your paper. Here's a polished summary of the differences between your work and existing approaches:
>
> First, in comparison to Retrieval-Augmented Generation (RAG), our \textsc{ReFeed} method also leverages retrieval as a crucial component of our pipeline. However, unlike conventional methods that directly use retrieval to enhance model performance in complex reasoning and factual accuracy, our research introduces a novel application. We demonstrate that retrieved documents can be effectively employed as feedback to refine language model outputs, significantly improving factual accuracy.
>
> Second, our approach diverges from recent advancements such as Chain-of-Thought, Tree-of-Thought reasoning, and Self-Refinement. These methods do not utilize external knowledge to augment language model reasoning and factual accuracy. Instead, they concentrate on improving the language model's reasoning abilities through diverse prompt designs.

---

> > ### Comment · Reviewer_fuhX · 2023-11-23
> > **Reply to Authors**
> >
> > Thank you for your answer.
> >
> > You have helped me to better understand.

---

### Official Review · Reviewer_Jo78 · 2023-10-31

**Soundness:** 3 good
**Presentation:** 2 fair
**Contribution:** 1 poor
**Rating:** 5
**Confidence:** 4

**Summary:**

This paper presents ReFeed, a retrieve-then-read pipeline for knowledge-intensive tasks. This pipeline involves three processes: generate the initial output only given the question, retrieve supporting documents using the question and the initial output, and finally refine the previous output. The authors further propose two enhancements: diversifying retrieval feedbacks and ensembling initial and post-feedback outputs.

**Strengths:**

The proposed method is straightforward and effective.

**Weaknesses:**

- The paper's novelty is in question, as the process of using an initial output for retrieval and then refining it does not appear to be a novel approach, and the absence of citation to related work, such as [1], raises concerns. Additionally, the paper lacks in-depth insights into the understanding of retrieve-then-read pipelines. The paper altogether seems more like a system report that shows the effectiveness of each trick, not an academic research.
- Typos
    - In Section 4.3.1, the second paragraph contains several sentences that are missing the subject.
    - In Section 4.3.2, the paragraph labeled "Module-1 …" mentions "as shown in 4", which should be corrected to "as shown in Figure 4”.

[1] Jiang et al. Active Retrieval Augmented Generation. 2023.

**Questions:**

How are the two enhanced modules integrated together?

---

> ### Author Response · Authors · 2023-11-23
> **Response to Reviewer Jo78**
>
> Dear Reviewer Jo78,
>
> Thanks very much for your review! Here are the responses to your questions/weakness.
>
> - Q1: The paper's novelty is in question, as the process of using an initial output for retrieval and then refining it does not appear to be a novel approach, and the absence of citation to related work, such as [1], raises concerns. Additionally, the paper lacks in-depth insights into the understanding of retrieve-then-read pipelines. The paper altogether seems more like a system report that shows the effectiveness of each trick, not an academic research.
>
> We acknowledge the concerns regarding the recent work by Jiang et al. on Active Retrieval Augmented Generation (2023). **It's important to note that our paper was published within 10 days on a third-party open-access repository. Due to the anonymity policy, we cannot specify the exact repository and date, but we emphasize that our work was indeed novel at the time of its publication and is contemporary with Jiang et al.'s research.**
>
> Despite the similarities in the pipeline (generate-retrieve-regenerate) between Active RAG and our ReFeed approach, we offer distinct contributions. Firstly, one-pass decoding in Active RAG may limit potential matches to diverse relevant information. Our research demonstrates that multi-pass decoding, or diverse generation, can facilitate more comprehensive retrieval feedback based on varying outputs.
>
> Secondly, we address a significant issue in the pipeline where retrieved documents might mislead the language model, leading to a correct output being inaccurately revised. We propose an ensemble strategy that allows for the re-evaluation of answer trustworthiness, thereby enhancing the reliability of the output. This aspect of our research presents a novel approach to managing the complexities inherent in retrieve-then-read pipelines.
>
> - Q2: How are the two enhanced modules integrated together?
>
> The two enhanced modules are built in two different steps. First, the diverse generation is built in the first step when generating initial answer, instead of generating only one answer, the system generates multiple diverse answer via sampling. Then, we use each generated initial answer to retrieve a set of documents and remove the duplicated documents. Lastly, the ensemble method is built in the second step when producing the final answer. The final output will be based on all initial answer likelihood and refined answer likelihood.
>
>
> - Q3: Typos.
>
> Thank you for pointing out the typos. We have revised them and updated the paper accordingly.

---

### Official Review · Reviewer_KK6a · 2023-10-31

**Soundness:** 3 good
**Presentation:** 3 good
**Contribution:** 3 good
**Rating:** 6
**Confidence:** 4

**Summary:**

This paper addresses factualness in QA systems. The idea of REFEED is as follow: 1) generate an output, 2) use retrieve relevant information from large document collections. 3) Integrate the latter into the prompt and refine the answer. The pipeline is plug-and-play and does not require any fine-tuning. The overall approach is more efficient and cost-effective than human feedback.

More in depths, REFEED generates multiple diverse outputs. For each, REFEED uses retrieval conditioned on the input and output instead of the input only, which makes it different than standard RAG approaches. The retriever (e.g., BM25) identifies the top-k relevant documents and remove duplicates. During refinement, the retrieved documents are integrated into the prompt along the outputs. However, there is a likelihood that the retrieved documents are misaligned with the output. The authors circumvent this problem by taking the average language modeling probability to rank the generated outputs before and after incorporating the retrieved documents. Then by comparing the different, the authors pick the first or refined output.

The experiments are based on single-hop QA, TriviaQA, multi-hop QA, and WoW. The models used are davinci-002 and davinci-003. The baselines are fair. The performance in the zero/few-shot experiments are convincing. The ablation study highlights the necessity of the multiple-output generation and the ensemble proposed to identify whether a refined output is better. Finally, the authors show how their approach can be integrated into chain-of-prompt and even improved the results.

Overall, this is a good paper, well written and structured. The idea, while simple, is novel. My only concern would be whether the proposed approach would work for other models than davinci-002/3 (see also related question regarding the calibration). I would ask the authors to experiment with one or two others LLMs (e.g., T5).

POST-REBUTTAL: Thank you for your answers. I will keep my current rating.

**Strengths:**

- Simple but effective method
- Strong results

**Weaknesses:**

- It is unclear whether the proposed approach would work with another backbone than davinci
- I'm skeptical that taking average probabilities of the output before and after the refinement would work in all models

**Questions:**

- How would verify that the LLM is well-calibrated in order to decide whether a refinement is more plausible or not? If the model is not well-calibrated, how would you proceed?
- How would the approach generalize for other models than davinci?

---

> ### Author Response · Authors · 2023-11-23
> **Response to Reviewer KK6a**
>
> Dear Reviewer KK6a,
>
> Thanks very much for your review! Here are the responses to your questions/weakness.
>
> - Q1: It is unclear whether the proposed approach would work with another backbone than davinci. How would the approach generalize for other models than davinci?
>
>
>
> Thank you for highlighting the importance of backbone model selection. Our initial decision to use Davinci-003 and Codex as backbone models was made to ensure a fair comparison with baseline methods (e.g., GenRead, RePLUG) that we have discussed in the paper.
>
> Besides, the ChatGPT and GPT-4 system might involve many user data for instruction tuning, which could have contamination with many academic benchmark dataset, such as NQ and TriviaQA.
>
> Lastly, in OpenAI’s announcement that both ChatGPT and GPT-4 will be subject to ongoing updates in their model parameters. These continual modifications would lead to non-reproducible experiments, potentially compromising the reliability of our research outcomes.
>
> In response to your concern, we have conducted additional experiments using ChatGPT (gpt-3.5 turbo) and GPT-4, which reinforce the adaptability and effectiveness of our approach across different model architectures. These results are now detailed in the revised version of our paper, providing a comprehensive view of our method's applicability.
>
>
> | Experiment on ChatGPT (gpt-3.5-turbo)                  | NQ          | TriviaQA    | HotpotQA    |
> |--------------------------------------------------------|-------------|-------------|-------------|
> | Closebook question answering (Question -> Answer)      | 32.3 / 39.9 | 65.6 / 69.5 | 23.5 / 24.0 |
> | Retrieve-then-Read (Question -> Retrieval -> Answer)   | 34.3 / 41.5 | 58.7 / 63.7 | 31.7 / 33.6 |
> | ReFeed (Question -> Answer -> Retrieval -> Refinement) | 37.5 / 48.1 | 66.3 / 71.1 | 34.1 / 36.0 |
>
>
> | Experiment on GPT4 (gpt-4)                             | NQ          | TriviaQA    | HotpotQA    |
> |--------------------------------------------------------|-------------|-------------|-------------|
> | Closebook question answering (Question -> Answer)      | 34.8 / 49.6 | 64.6 / 72.8 | 30.8 / 33.7 |
> | Retrieve-then-Read (Question -> Retrieval -> Answer)   | 32.5 / 46.5 | 59.9 / 67.1 | 31.6 / 37.5 |
> | ReFeed (Question -> Answer -> Retrieval -> Refinement) | 36.8 / 54.4 | 66.3 / 74.0 | 36.9 / 42.6 |
>
> From the above table, we can obserbe:
>
> (1) The retrieval-feedback consistently improves the system, compared to both standard closebook QA method, and retrieve-then-read pipeline.
>
> (2) The traditional retrieve-then-read system even sometimes hurt by noisy retrieval. On the contrary, our ReFeed could avoid misled issue if the first-round generation is good.
>
>
> - Q2: I'm skeptical that taking average probabilities of the output before and after the refinement would work in all models. How would verify that the LLM is well-calibrated in order to decide whether a refinement is more plausible or not? If the model is not well-calibrated, how would you proceed?
>
> Thanks for your question! Indeed, there is a fundamental limitation in ensuring that the language model is appropriately calibrated. And we admit our method should be based on a relatively well-calibrated language model.
>
> If the model is not well-calibrated, one way is to better measure uncertainty using semantic-based metrics [1], instead of using probability only.
>
> We add a limitation discussion in the paper to point out the issue and remind our ensemble strategy could only be applied on well-calibrated model. Besides that, our main retrieval feedback framework is not relying on model calibration.

---

### Official Review · Reviewer_N6CA · 2023-11-04

**Soundness:** 3 good
**Presentation:** 2 fair
**Contribution:** 3 good
**Rating:** 6
**Confidence:** 5

**Summary:**

In this paper, the authors propose to leverage the outputs of LLMs (i.e., the generated initial answers) to retrieve relevant documents for refining LLM outputs and quality. They also present two modules to enhance the performance by diversifying retrieval feedback and ensembling initial and post-feedback outputs. Experiments are conducted on several conventional QA benchmark datasets. The experimental results demonstrate the proposed method can improve the performance of different LLMs in both close- and open-book settings. The authors also conducted some ablation studies to show the effectiveness of each component.

**Strengths:**

* S1: The proposed framework is an easy-to-use and plug-and-play blackbox retrieval-augment approach.
* S2: The improvements over baseline methods are significant across different datasets and settings.
* S3: Each proposed component is validated through the ablation study

**Weaknesses:**

* W1: Some important details are missing, e.g., how to conduct de-duplication;
* W2: Not applied to the state-of-the-art LLMs (e.g., GPT-4).
* W3:Lack of discussions and analysis on how hyper-parameters affect the performance.

**Questions:**

* Q1: Following W1, I would encourage the authors to describe the methods with details and motivation, especially when each component is shown effective individually. For instance, I wonder how the de-duplication is done to ensure diversity; and why BM25 is chosen instead of other retrieval methods (of course not just answering "other methods also just used it").

*Q2: Following W2, I would really like to know how the proposed method can be applied to state-of-the-art models like GPT-4. Although it might not be reproducible for a certain metric number, it can still give some signs about the performance upper bound. Similarly, I wonder if the method can be applied to a conventional smaller neural language model (e.g., BART and T5) since the proposed is a plug-and-play method.

*Q3: Following W3, there are still several hyper-parameters (e.g., k in the module-1) in the proposed method. I wonder how they affect the performance (instead of "because other papers use this number").

**Details Of Ethics Concerns:**

N/A.

---

> ### Author Response · Authors · 2023-11-23
> **Response to Reviewer N6CA [1/2]**
>
> Dear Reviewer N6CA,
>
> Thanks very much for your review! Here are the responses to your questions/weakness.
>
> - Q1: Following W1, I would encourage the authors to describe the methods with details and motivation, especially when each component is shown effective individually. For instance, I wonder how the de-duplication is done to ensure diversity; and why BM25 is chosen instead of other retrieval methods (of course not just answering "other methods also just used it").
>
> We appreciate your emphasis on the need for detailed methodological descriptions. Regarding the de-duplication process, we first use each generated initial answer to retrieve a set of documents. For each document retrieved, we calculate a corresponding similarity score. Subsequently, we merge all retrieved documents and rank them based on their similarity scores, keeping only the top-k documents. This approach ensures that we maintain diversity in the information we gather and use.
>
> There are two main reasons for choosing BM25 as our retrieval method. First, BM25 demonstrates excellent adaptability and generalizability across a variety of contexts. Second, dense retrieval methods, such as DPR, are often trained with QA datasets or synthetic query-document pairs. Therefore, evaluating such dense retrieval methods might disproportionately reflect their ability to provide retrieval feedback, as the retrieval more accurate than BM25. However, demonstrating that BM25 performs well within our retrieval feedback framework suggests that there is potential for significant improvement with more advanced retrieval methods.
>
>
> Lastly, we want to summarize out motivation of each component:
>
> (1)	General Retrieval Feedback Pipeline: Language models often generate information that is either incorrect or hallucinated. The retrieved information allows the language model to reassess the initial outputs and, if necessary, refine them to produce new answers.
>
> (2)	The diverse retrieval strategy enhances the diversity of the outputs and enables more comprehensive retrieval feedback based on these varied outputs.
>
> (3)	Ensemble method is motivated that there are times when the retrieved documents may mislead the language model, leading to a correct output being revised into an incorrect one. So, the ensemble technique takes into account both the initial and refined outputs, ultimately enhancing the overall performance.
>
> - Q2: Following W2, I would really like to know how the proposed method can be applied to state-of-the-art models like GPT-4. Although it might not be reproducible for a certain metric number, it can still give some signs about the performance upper bound. Similarly, I wonder if the method can be applied to a conventional smaller neural language model (e.g., BART and T5) since the proposed is a plug-and-play method.
>
>
> Thank you for highlighting the importance of backbone model selection. Our initial decision to use Davinci-003 and Codex as backbone models was made to ensure a fair comparison with baseline methods (e.g., GenRead, RePLUG) that we have discussed in the paper.
>
> Besides, the ChatGPT and GPT-4 system might involve many user data for instruction tuning, which could have contamination with many academic benchmark dataset, such as NQ and TriviaQA.
>
> Lastly, in OpenAI’s announcement that both ChatGPT and GPT-4 will be subject to ongoing updates in their model parameters. These continual modifications would lead to non-reproducible experiments, potentially compromising the reliability of our research outcomes.
>
> In response to your concern, we have conducted additional experiments using ChatGPT (gpt-3.5 turbo) and GPT-4, which reinforce the adaptability and effectiveness of our approach across different model architectures. These results are now detailed in the revised version of our paper, providing a comprehensive view of our method's applicability.
>
>
> | Experiment on ChatGPT (gpt-3.5-turbo)                  | NQ          | TriviaQA    | HotpotQA    |
> |--------------------------------------------------------|-------------|-------------|-------------|
> | Closebook question answering (Question -> Answer)      | 32.3 / 39.9 | 65.6 / 69.5 | 23.5 / 24.0 |
> | Retrieve-then-Read (Question -> Retrieval -> Answer)   | 34.3 / 41.5 | 58.7 / 63.7 | 31.7 / 33.6 |
> | ReFeed (Question -> Answer -> Retrieval -> Refinement) | 37.5 / 48.1 | 66.3 / 71.1 | 34.1 / 36.0 |
>
>
> | Experiment on GPT4 (gpt-4)                             | NQ          | TriviaQA    | HotpotQA    |
> |--------------------------------------------------------|-------------|-------------|-------------|
> | Closebook question answering (Question -> Answer)      | 34.8 / 49.6 | 64.6 / 72.8 | 30.8 / 33.7 |
> | Retrieve-then-Read (Question -> Retrieval -> Answer)   | 32.5 / 46.5 | 59.9 / 67.1 | 31.6 / 37.5 |
> | ReFeed (Question -> Answer -> Retrieval -> Refinement) | 36.8 / 54.4 | 66.3 / 74.0 | 36.9 / 42.6 |

---

> ### Author Response · Authors · 2023-11-23
> **Response to Reviewer N6CA [2/2]**
>
> From the above table, we can obserbe:
>
> (1) The retrieval-feedback consistently improves the system, compared to both standard closebook QA method, and retrieve-then-read pipeline.
>
> (2) The traditional retrieve-then-read system even sometimes hurt by noisy retrieval. On the contrary, our ReFeed could avoid misled issue if the first-round generation is good.
>
>
>
> - Q3: Following W3, there are still several hyper-parameters (e.g., k in the module-1) in the proposed method. I wonder how they affect the performance (instead of "because other papers use this number").
>
> Previous research, as mentioned in papers [1] and [2], indicates that increasing the value of 'k' beyond 10 leads to only marginal improvements while simultaneously contributing to higher complexity. Additionally, as noted in [3], expanding the context length excessively for LLMs can result in “lost in the middle issue”, i.e., LLM ignores the middle part of a long text input.
>
> To further investigate the effect of varying 'k', we conducted experiments using ChatGPT (gpt-3.5-turbo) on datasets such as NQ, TriviaQA, and HotpotQA. The results are shown in the following tables (metric: left EM, right accuracy). The experimental results are consistent with findings in [1] [2]. As shown in the Table when ‘k’ is set as 10 performs significantly better than when ‘k’ is set as 1 or 5, however, when ‘k’ is set as 20, the improvement is marginal over when ‘k’ is set as 10.
>
> We also observe that when 'k' is set as a small number, such as 1, the language model could be misled because it tends to trust the only one given documents.
>
> | K=1 with ChatGPT (gpt-3.5-turbo)                       | NQ          | TriviaQA    | HotpotQA    |
> |--------------------------------------------------------|-------------|-------------|-------------|
> | Closebook question answering (Question -> Answer)      | 32.3 / 39.9 | 65.6 / 69.5 | 23.5 / 24.0 |
> | Retrieve-then-Read (Question -> Retrieval -> Answer)   | 23.6 / 30.0 | 40.5 / 44.8 | 23.2 / 24.0 |
> | ReFeed (Question -> Answer -> Retrieval -> Refinement) | 34.0 / 41.9 | 65.9 / 68.7 | 27.1 / 28.3 |
>
> | K=5 with ChatGPT (gpt-3.5-turbo)                       | NQ          | TriviaQA    | HotpotQA    |
> |--------------------------------------------------------|-------------|-------------|-------------|
> | Closebook question answering (Question -> Answer)      | 32.3 / 39.9 | 65.6 / 69.5 | 23.5 / 24.0 |
> | Retrieve-then-Read (Question -> Retrieval -> Answer)   | 30.5 / 37.6 | 50.4 / 55.4 | 28.5 / 29.0 |
> | ReFeed (Question -> Answer -> Retrieval -> Refinement) | 34.3 / 42.1 | 67.5 / 70.8 | 30.0 / 30.4 |
>
> | K=10 with ChatGPT (gpt-3.5-turbo)                      | NQ          | TriviaQA    | HotpotQA    |
> |--------------------------------------------------------|-------------|-------------|-------------|
> | Closebook question answering (Question -> Answer)      | 32.3 / 39.9 | 65.6 / 69.5 | 23.5 / 24.0 |
> | Retrieve-then-Read (Question -> Retrieval -> Answer)   | 34.3 / 41.5 | 58.7 / 63.7 | 31.7 / 33.6 |
> | ReFeed (Question -> Answer -> Retrieval -> Refinement) | 37.5 / 48.1 | 66.3 / 71.1 | 34.1 / 36.0 |
>
> | K=20 with ChatGPT (gpt-3.5-turbo)                      | NQ          | TriviaQA    | HotpotQA    |
> |--------------------------------------------------------|-------------|-------------|-------------|
> | Closebook question answering (Question -> Answer)      | 32.3 / 39.9 | 65.6 / 69.5 | 23.5 / 24.0 |
> | Retrieve-then-Read (Question -> Retrieval -> Answer)   | 34.6 / 42.0 | 59.5 / 64.5 | 31.8 / 33.8 |
> | ReFeed (Question -> Answer -> Retrieval -> Refinement) | 37.8 / 48.2 | 66.5 / 71.2 | 34.4 / 36.5 |
>
>
> [1] Prompting GPT-3 To Be Reliable. ICLR 2023
>
> [2] REPLUG: Retrieval-Augmented Black-Box Language Models. EMNLP 2023.
>
> [3] Lost in the Middle: How Language Models Use Long Contexts. TACL 2023.

---

> ### Comment · Reviewer_N6CA · 2023-11-23
>
> I acknowledge that I have read all of the author responses.

---

### Author Response · Authors · 2023-11-23
**General Response**

Dear Reviewers and Area Chair,

Thank you very much for reviewing our paper! Your professional reviews offer us great advice towards writing a more comprehensive paper! And, we are pleased that most reviewers appreciated the contributions of our work. To summarize our contribution, we present a novel pipeline designed to enhance LLMs by providing automatic retrieval feedback in a plug-and-play framework without the need for expensive fine-tuning.

Following the rebuttal period, we have incorporated several significant changes into our paper:

- On [Page 8, Table 6], we have added experiments conducted on state-of-the-art GPT models, including ChatGPT-3.5 and GPT-4.
- [Page 9] now includes a detailed discussion on the limitations of our study.
- We expanded the discussion regarding hyper-parameter settings and implementation details on [Page 13].
- [Page 14] features experiments involving the sampling of different numbers of documents (exploring various choices of k).
- Additionally, [Page 3] has been enriched with an extended discussion on related work, including CoT, ToT, and RAG.

We believe these revisions have substantially enhanced the quality and depth of our paper. We extend our sincerest thanks to the reviewers for their valuable suggestions and feedback.

Should you find that our revisions satisfactorily address your concerns, we would be grateful if you would consider raising the score of our submission.

Best regards,

ReFeed authors

---

### Meta-Review · Area_Chair_4ZGY · 2023-12-05

**Metareview:**

The submission introduces REFEED, a framework leveraging language model outputs to refine answers via document retrieval. Reviewers generally commend the simplicity and effectiveness of REFEED's architecture, highlighting its significant improvements over baselines across diverse QA datasets. However, concerns persist regarding novelty, missing details (e.g., deduplication), and limited exploration beyond specific language models. The rebuttals partially address concerns on novelty and reliability enhancement but lack a comprehensive analysis of underlying principles. Some reviewers question the clarity of goals, absence of comparisons with other approaches like RAG, and incomplete reproducibility details. While appreciating author responses, reviewers remain skeptical about novelty and effectiveness, suggesting the need for further revisions to address these concerns. The paper is seen as a compilation of empirical techniques lacking in-depth analysis of its core principles, resulting in a recommendation for a weak rejection.

**Justification For Why Not Higher Score:**

The collective feedback indicates that significant improvements and clarifications are needed to enhance the paper's novelty, rigor, and comprehensiveness to warrant a higher score.

**Justification For Why Not Lower Score:**

N/A

---

### Decision · Program_Chairs · 2024-01-16

Reject